

# Monitoring continuous spectrum observables: the strong measurement limit

**Michel Bauer[1,2], Denis Bernard[3*] and Tony Jin[3]**

**1** Institut de Physique Théorique de Saclay, CEA-Saclay & CNRS,
91191 Gif-sur-Yvette, France.
**2** Département de mathématiques et applications, École Normale Supérieure,
PSL Research University, 75005 Paris, France
**3** Laboratoire de Physique Théorique de l'École Normale Supérieure de Paris,
CNRS, ENS, PSL University & Sorbonne Université, France.

* denis.bernard@ens.fr

## Abstract

We revisit aspects of monitoring observables with continuous spectrum in a quantum system subject to dissipative (Lindbladian) or conservative (Hamiltonian) evolutions. After recalling some of the salient features of the case of pure monitoring, we deal with the case when monitoring is in competition with a Lindbladian evolution. We show that the strong measurement limit leads to a diffusion on the spectrum of the observable. For the case with competition between observation and Hamiltonian dynamics, we exhibit a scaling limit in which the crossover between the classical regime and a diffusive regime can be analyzed in details.

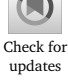

# 1  Introduction

Upon monitoring a quantum system, one progressively extracts random information and back-acts randomly on the system [1–3]. There are of course various way to monitor a quantum system [4, 5]. Let us assume that the observation process is such that if the monitoring is strong enough the result of these back-actions is to project the system state close to one of the eigenstates of the monitored observable. Then, if the monitoring process and the system dynamics are compatible – that is, if the flows they induce are commuting –, the net effect of the combined system dynamics plus monitoring is this projection mechanism. However if the system dynamics is not compatible with the monitoring process, the resulting dynamical evolution is more subtle and depends on the relative competition between these two processes.

In the case of finite dimensional systems, the effective dynamics is simple enough. The observable eigenstates are well-defined normalisable states, often called pointer states [6]. In absence of system dynamics, strongly monitoring the system results in non-linearly projecting its state onto one of the pointer state according to Born's rules. This is usually referred to as non-demolition measurement [4, 5, 7]. Adding a non trivial system dynamics, be it coherent or dissipative, induces quantum jumps between pointer states [8–11]. These jumps occurs randomly and are governed by a Markov chain whose transition rates depend on the system dynamics and the nature of the monitored observable [12–14]. These jumps processes are dressed by quantum spikes but we shall here neglect these finer effects [15, 16]. One noticeable difference between coherent and dissipative system dynamics is that the former is Zeno frozen [17] under monitoring so that the transition rates of the quantum jump Markov chain asymptotically depend on the strength of the monitoring process while they do not in the latter case.

The picture is a priori more complicated in the case of the monitoring of an observable with continuous spectrum [18–21], because then the observable eigenstates are not normalisable – they are generalised eigenstates –, so that the result of the monitoring process cannot be the projection of the system state onto one of those non-normalisable states and the result of the combined system plus monitoring dynamics cannot be a simple Markov chain on those generalised eigenstates.

The dynamical process and the statistical aspects involved in monitoring an observable with continuous spectrum has recently been precisely described in ref. [22], in the absence of internal dynamics. The aim of this paper is to show that, at least for some classes of system dynamics, one can present a precise formulation of the resulting stochastic processes describing the combined system plus observation dynamics in the limit of strong monitoring of an observable with continuous spectrum. Naively, we may expect that those processes are some kind of diffusion processes on the spectrum of the monitored observable. The aim of this paper is to make this statement more precise. To fix the framework, we shall assume that the monitored observable is the position observable for a massive particle on flat space, but the generalisation to other observables is clear. We shall deal with dissipative and hamiltonian dynamics. In both the Lindbladian and the Hamiltonian case we start from an exact treatment of a simple model (related in both cases, though for different reasons, to an harmonic oscillator)

and then extend our results to more general situations.

The case of dissipative dynamics is dealt with in Section 3. There we shall prove that, in the limit of strong monitoring of the position observable, the resulting processes are driven by stochastic differential equations on the position space, if the Lindblad operators generating the system dynamics are quadratic in the impulsion operators. In other words, under such hypothesis, quantum trajectories are mapped into stochastic trajectories – a kind of quantum to classical transition induced by monitoring.

The case of hamiltonian dynamics is dealt with in Section 4. There we take a look at the well-documented problem of a quantum massive particle in a smooth potential whose position is monitored continuously [18–21, 23–25]. As previously analysed [23, 26–28, 30], in such a case the resulting dynamics possesses three different regimes: (a) a collapse regime in which the wave function localizes in space in agreement with non-demolition measurement, (b) a classical regime in which the localized wave function moves in space according to classical dynamics, and (c) a diffusive regime in which the wave function diffuses randomly and significantly. However, as pointed out in ref. [30], "it is not an easy task to spell out rigorously these regimes and their properties". The aim of this section is to argue that we may define a double scaling limit which maps quantum trajectories onto solutions of a Langevin equation describing precisely enough the cross-over from the classical regime to the diffusive regime.

How to formulate position monitoring in quantum mechanics and what is the associated notion of quantum trajectories is recalled in the Section 2. This Section is also used to describe fine statistical aspects of position monitoring. In particular we explain how one may adopt two different points of view to describe those statistical properties depending on which information one has access to. Some mathematical results related to this last point – i.e. related to possible changes of filtrations in observing the monitoring process – are given in the Appendix.

## 2 QND measurement of a continuous spectrum observable.

The general rules of quantum mechanics (including the measurement postulates) enable to show that the evolution of the density matrix under continuous monitoring of the observable $X$ (later we shall take $X$ to be the position observable, hence the name) but in absence of any internal evolution read [18–21] (see the lecture notes [29] for a simple derivation):

$$
\begin{aligned}
d\rho_t &= -\frac{\gamma^2}{2}[X,[X,\rho_t]]dt + \gamma\big(X\rho_t + \rho_t X - 2\langle X\rangle_t \rho_t\big)dW_t, \\
dS_t &= 2\gamma\langle X\rangle_t\, dt + dW_t,
\end{aligned}
\tag{1}
$$

with $\langle X\rangle_t := \mathrm{Tr}(X\rho_t)$ and $W_t$ a normalized Brownian motion, with $dW_t^2 = dt$, and $S_t$ the output signal. The dimensionful parameter $\gamma$ codes for the rate at which information is extracted. One may alternatively write the above evolution equation eq.(1) for the density matrix kernel $\rho_t(x,y) = \langle x|\rho_t|y\rangle$, with $|x\rangle$ the generalized eigenstate of $X$ with eigenvalue $x$, as:

$$
d\rho_t(x,y) = -\frac{\gamma^2}{2}(x-y)^2\rho_t(x,y)dt + \gamma\big((x+y-2\langle X\rangle_t)\rho_t(x,y)\big)dW_t,
\tag{2}
$$

with $\langle X\rangle_t := \int \mathrm{d}x\, x\rho_t(x,x)$.

One way to derive these equations is to look at a discrete time version where the monitoring is the effect of repeated interactions of probes with the system, with subsequent "yes-no" von Neumann measurement coupled to $X$ on the probes. The process $S$ is the bookkeeping device to record the monitoring and in the discrete-time viewpoint, a "yes" (resp. a "no") for a probe measurement leads to an increase (resp. decrease) of $S$ for the corresponding time step. Probe measurements with more than two possible outcomes lead to introduce vectors $S$ and $W$.

The purpose of the next subsection is to exhibit two equivalent, but rather different looking, descriptions of the monitoring process. We state the main results, but the details are relegated to the appendix, together with a quick discussion of the discrete time situation which the reader should refer to for motivation.

## 2.1 Diagonal evolution and probabilistic interpretation

The point of view we develop now is the following: during the monitoring process of the observable $X$, the observer gains information little by little and is able at large times to infer a value, say $x$, that characterizes the asymptotic state of the system. It turns out that the description of the process is substantially simpler for a "cheater" who uses from the beginning the knowledge of the result of the measurement. Though we do not claim that this is "the" interpretation of the density matrix, it is convenient for the forthcoming discussion to talk about it (in particular about its diagonal with respect to the spectral decomposition of $X$) using a Bayesian vocabulary, i.e. view it as expressing the best guess for the observer with the information available for him at a certain time.

If $X$ is (as suggested by the name) the position of the particle, the "eigenstates" of $X$ are not normalizable, so that the matrix elements of the density matrix between those "eigenstates", also called pointer states, may not be well-defined: in general the diagonal of the density matrix is not a function of the position variable $x$ but a probability measure $d\mu_t(x)$. In terms of the density matrix kernel we have $d\mu_t(x) = \rho_t(x, x) dx$.

We choose to concentrate on the diagonal of the density matrix in the pointer states basis for the forthcoming discussion. Happily, the time evolution of the pair $(d\mu_t, S_t)_{t \geq 0}$, which is easily deduced from the general equations above, eqs.(1,2), remains autonomous and reads [1]:

$$d\,d\mu_t(x) = 2\gamma\left(x - \int y\,d\mu_t(y)\right)d\mu_t(x)\,dW_t, \tag{3}$$

$$dS_t = 2\gamma\left(\int y\,d\mu_t(y)\right)dt + dW_t. \tag{4}$$

The measure $d\mu_0$, i.e. the diagonal of the density matrix before the monitoring starts, is assumed to be known to the experimenter (e.g. via preparation). We shall see that $d\mu_t$ can be interpreted as a best guess at time $t$ for the experimenter knowing only $d\mu_0$ and $S_u, u \in [0, t]$.

To simplify the notation a little bit, we set $A := 2\gamma X$ and use $\alpha$ to denote the variables in the spectrum of $A$. The above coupled stochastic differential equations are not totally trivial to analyze. However, the following holds:

**Proposition:** *Let $(B_t)_{t \geq 0}$ be a Brownian motion and $A$ be an independent random variable with distribution $d\mu_0(\alpha)$ defined on some common probability space.*
*Let $S_t$ be the process $S_t := B_t + At$. Let $\mathcal{G}_t := \sigma\{A \text{ and } S_u, u \in [0, t]\} = \sigma\{A \text{ and } B_u, u \in [0, t]\}$ be the filtration describing the knowledge accumulated by knowing $A$ since $t = 0$ and $B_u$ or $S_u$ for $u \in [0, t]$. Let $\mathcal{H}_t := \sigma\{S_u, u \in [0, t]\}$ be the filtration describing the knowledge accumulated by knowing $S_u$ for $u \in [0, t]$. Then:*
*– If $h$ is an arbitrary measurable function such that $h(A)$ is integrable then*

$$\mathbb{E}[h(A)|\mathcal{H}_t] = \frac{\int d\mu_0(\beta)h(\beta)e^{\beta S_t - \beta^2 t/2}}{\int d\mu_0(\beta)e^{\beta S_t - \beta^2 t/2}}.$$

*– The process $(W_t)_{t \geq 0}$ defined by*

$$W_t := S_t - \int_0^t \mathbb{E}[A|\mathcal{H}_u]\,du$$

---

[1]Don't get confused between the notation $d$ associated to the integration of the variable $\alpha$ or $x$ and the notation $d$ related to differentiation with respect to time variable $t$.

*is an $\mathcal{H}_t$-adapted Brownian motion.*

*– The pair $(d\mu_t, S_t)_{t \geq 0}$ where $\mu_t$ is defined by*

$$\mathbb{E}[h(A)|\mathcal{H}_t] =: \int d\mu_t(\beta)h(\beta), \quad i.e. \ d\mu_t(\alpha) = \frac{d\mu_0(\alpha)e^{\alpha S_t - \alpha^2 t/2}}{\int d\mu_0(\beta)e^{\beta S_t - \beta^2 t/2}} \tag{5}$$

*solves the system*

$$dd\mu_t(\alpha) = \left(\alpha - \int \beta \, d\mu_t(\beta)\right)d\mu_t(\alpha) \, dW_t, \tag{6}$$

$$dS_t = \left(\int \beta \, d\mu_t(\beta)\right)dt + dW_t, \tag{7}$$

A detailed proof in given in the appendix, together with some motivations from the discrete time situation. The proof we give is a pedestrian one, based on explicit translations in Gaussian integrals. This is also the main intuition behind Girsanov's theorem, which is indeed the right general framework in which the proposition fits.

Let us now explain the meaning of this result. The important thing to realize is that the process $S_t$ can be analyzed under a number of filtrations, but its definition and properties are independent of the choice of filtration.

For instance, from the initial decomposition $S_t = B_t + At$ we infer that $\frac{S_t}{t}$ converges at large times to the random variable $A$ with probability 1, because by the law of large numbers for Brownian motion $\frac{B_t}{t}$ converges at large times to 0 with probability 1. By the law of large numbers for Brownian motion again $\frac{W_t}{t}$ converges at large times to 0 with probability 1. Hence, the $\mathcal{H}_t$-measurable random variable $\frac{1}{t}\int_0^t \mathbb{E}[A|\mathcal{H}_u] \, du$ which, by eq.(5), equals $\frac{1}{t}\int_0^t \left(\int \beta \, d\mu_u(\beta)\right) du$ converges at large times to the random variable $A$ with probability 1.

Now $\mathbb{E}[A|\mathcal{H}_u]$ is, by the very definition of conditional expectations, the best $\mathcal{H}_u$-measurable approximation of $A$. Thus the connection between the definition $S_t = B_t + At$ and the system of eqs.(6,7) has an interpretation in the field of statistical estimation: It is equivalent to sample $A$ at time 0 (with $d\mu_0$ of course) and observe the process $(S_t)_{t \geq 0}$ (i.e. to use the filtration $\mathcal{G}_t$) or to observe only $(S_t)_{t \geq 0}$ (i.e. to use the filtration $\mathcal{H}_t$) and retrieve $A$ asymptotically using Bayesian inference. Another road to this result is to substitute $S_t = B_t + At$ in the formula for $d\mu_t(\alpha)$ to find that the numerator $e^{\alpha S_t - \alpha^2 t/2}$ is strongly peaked around $\alpha = A$ at large times.

The striking fact is that the systems (3,4) and (6,7) are the same (mutatis mutandi, with the substitution $A = 2\gamma X$). But the first system results from the application of the rules of quantum mechanics, while the second one has a purely statistical estimates content as explained above. From the point of view of quantum mechanics, the natural situation of an experimenter is that she/he observes only the result of monitoring i.e. the process $(S_t)_{t \geq 0}$ and infers/measures a more and more accurate value of $X$ when time gets large. But there is also an interpretation when a cheater measures $X$ at time 0 (with the outcome distributed as $d\mu_0$ of course) and then gives the system to the experimenter. The cheater has $\mathcal{G}_t$ at its disposal, and in particular knows in advance which value for $X$ the experimenter will infer/measure after an infinite time from the sole result of monitoring.

The explicit formula for $d\mu_t$ as a function of $S_t$ can be quickly recovered by linearization of the system (6,7), a trick that we recall below because it works in general (i.e. when the system has some intrinsic evolution while the monitoring is performed). Note also that $W_t$ is not simply the conditional expectation of $B_t$ knowing $\mathcal{H}_t$. The interested reader should consult the appendix for a detailed discussion.

## 2.2 Density matrix evolution

Let us now give a brief description of the off-diagonal elements of the density matrix kernel $\rho_t(x, y) = \langle x | \rho_t | y \rangle$ whose evolution is governed by eq.(2). These simple results will be useful in the following. As is well known [23–25], the solution of this equation are obtained by changing variables and defining a un-normalized density matrix kernel $\hat{\rho}_t(x, y)$, solution of the linear stochastic differential equation

$$d\hat{\rho}_t(x, y) = -\frac{\gamma^2}{2}(x-y)^2 \hat{\rho}_t(x, y) dt + \gamma(x+y)\hat{\rho}_t(x, y)\big) dS_t, \qquad (8)$$

driven by the output signal $S_t$. The normalized density kernel $\rho_t(x, y)$ is then reconstructed by setting

$$\rho_t(x, y) = \frac{\hat{\rho}_t(x, y)}{Z_t},$$

with the normalization factor $Z_t := \int dx \, \hat{\rho}_t(x, x)$ satisfying $dZ_t = 2\gamma \langle X \rangle_t Z_t \, dS_t$. Since $dS_t^2 = dt$, solution of eq.(8) reads

$$
\begin{aligned}
\hat{\rho}_t(x, y) &= \rho_0(x, y) \, e^{-\gamma^2 t (x^2 + y^2) + \gamma(x+y)S_t}. \\
Z_t &= \int d\mu_0(x) \, e^{-2\gamma^2 t x^2 + 2\gamma x S_t}.
\end{aligned}
$$

For the diagonal elements one recovers the solution (5) described above. The mean position is then $2\gamma \langle X \rangle_t = (\partial_S \log Z)(S_t)$ so that $dS_t = (\partial_S \log Z)(S_t) dt + dW_t$.

The analysis of the previous subsection implies that the signal $S_t$ can also be decomposed as $S_t = 2\gamma \bar{x} t + B_t$, with $B_t$ another standard Brownian motion and $\bar{x}$ an independent random variable sampled with measure $d\mu_0(x)$. As a consequence,

$$\hat{\rho}_t(x, y) = \rho_0(x, y) e^{2\gamma^2 t \bar{x}^2} \, e^{-\gamma^2 t \left((x-\bar{x})^2 + (y-\bar{x})^2\right) + O(\gamma\sqrt{t})},$$

and $d\mu_t(x)$ is a Gaussian distribution[2] centred at the random position $\bar{x}$ with width of order $\sim 1/\sqrt{\gamma^2 t}$. Alternatively, this tells us that under QND monitoring of the position the system state has approximatively 'collapses' on Gaussian state of width of order $\ell$ after a time of order $1/\ell^2 \gamma^2$.

The diagonal components of the density matrix kernel have a well-defined large $\gamma$ limit, in the sense that $d\mu_t(x) \to \delta(x - \bar{x})dx$ in this limit, but the off-diagonal components don't. This is illustrated by the behaviour of the momentum distribution, $\langle e^{iaP} \rangle_t = e^{-\frac{1}{2}\gamma^2 t a^2}$, which implies that $\langle P^{2k} \rangle_t \to \infty$ and $\langle f(P) \rangle_t \to 0$ for any function $f$ with bounded Fourier transform, at large $\gamma$ as expected from Heisenberg principle.

# 3 Monitoring continuous spectrum observables with dissipative dynamics

The aim of this section is to understand what dynamical processes emerge when strongly monitoring an observable with a continuous spectrum for a quantum system subject to a dissipative dynamics.

The simplest way to present the discussion is to consider a quantum particle on the real line, with quantum Hilbert space $\mathcal{H} = L^2(\mathbb{R})$, and a monitoring of the position observable $X$.

---

[2]But the kernel $\rho_t(x, y)$ are not a two dimensional normalized Gaussian.

The (stochastic) dynamical equation for the density matrix is then

$$d\rho_t = L(\rho_t)\,dt - \frac{\gamma^2}{2}[X,[X,\rho_t]]\,dt + \gamma\big(X\rho_t + \rho_t X - 2\langle X\rangle_t \rho_t\big)\,dW_t, \qquad (9)$$

with $W_t$ a normalized Brownian motion and $L$ some Lindblad operator. Alternatively, the evolution of the density kernel $\rho_t(x,y) = \langle x|\rho_t|y\rangle$ reads

$$d\rho_t(x,y) = L(\rho_t)(x,y)\,dt - \frac{\gamma^2}{2}(x-y)^2\rho_t(x,y)\,dt + \gamma\big((x+y-2\langle X\rangle_t)\rho_t(x,y)\big)\,dW_t.$$

The aim of this section is to understand the limit of large $\gamma$.

If one want to get a precise statement, one should not look directly at the dynamics of the limiting density kernel but at the dynamics of the measures associated to system observables induced by the density matrix. Let us recall that, given a system density matrix $\rho$, to any system observable $O = O^\dagger$ is associated a measure $\mu^O$ on $Spec(O)$, the spectrum of $O$, via

$$\mathrm{Tr}\big(\rho\,\varphi(O)\big) = \int d\mu^O[o]\,\varphi(o),$$

for any (bounded) function $\varphi$. Now, a time evolution $\rho \to \rho_t$, as specified by eq.(9), induces a time evolution of the measure $\mu^O \to \mu^O_t$ via $\int d\mu^O_t[o]\,\varphi(o) = \mathrm{Tr}\big(\rho_t\,\varphi(O)\big)$. Since $\rho_t$ is random, so is $\mu^O_t$. The statements about the large $\gamma$ limit are going to be formulated in terms of the limiting behavior of those measures for appropriately chosen observables.

The measure we should look at is that associated to the position, the observable that is monitored, defined by

$$\int d\mu^X_t(x)\,\varphi(x) = \mathrm{Tr}\big(\rho_t\,\varphi(X)\big),$$

for any function $\varphi$. To simplify notation, we drop the upper index $X$ on $\mu^X$ and let $d\mu_t(x) := d\mu^X_t(x)$. Alternatively, $d\mu_t(x) = \rho_t(x,x)\,dx$.

## 3.1 A simple example: Monitoring quantum diffusion.

Let us first look at the simplest model for which the Lindblad operator $L$ is the so-called quantum Laplacian

$$L(\rho) = -\frac{D}{2}\big[P,[P,\rho]\big],$$

with $D$ some diffusion constant, so that $L(\rho_t)(x,y) = \frac{D}{2}(\partial_x + \partial_y)^2\rho_t(x,y)$. With this choice, the SDE for the density kernel reads

$$d\rho_t(x,y) = \frac{D}{2}(\partial_x + \partial_y)^2\rho_t(x,y)\,dt - \frac{\gamma^2}{2}(x-y)^2\rho_t(x,y)\,dt + \gamma\big((x+y-2\langle X\rangle_t)\rho_t(x,y)\big)\,dW_t. \qquad (10)$$

Interestingly enough, this equation is closed on the diagonal elements so that the measure $d\mu_t(x) := \rho_t(x,x)\,dx$ satisfies

$$d\big(d\mu_t(x)\big) = \frac{D}{2}\partial_x^2\,d\mu_t(x)\,dt + 2\gamma\,(x - \langle X\rangle_t)\,d\mu_t(x)\,dW_t,$$

with $\langle X\rangle_t = \int d\mu_t(x)\,x =: \mu_t[x]$. Alternatively, introducing a test function $\varphi$ and denoting its "moment" by $\mu_t[\varphi] = \int d\mu_t(x)\,\varphi(x)$, the stochastic evolution reads:

$$d\,\mu_t[\varphi] = \frac{D}{2}\mu_t[\Delta\varphi]\,dt + 2\gamma\,\mu_t[x\cdot\varphi]^c\,dW_t, \qquad (11)$$

with $\Delta = \partial_x^2$ the Laplacian, and $\mu_t[x \cdot \varphi]^c := \mu_t[x\varphi] - \mu_t[x]\mu_t[\varphi]$ the "connected $[x \cdot \varphi]$-moment".

Eq.(11) defines a process on measures on the real line. It is specified by its transition kernel which is going to be generated by a second order differential operator. We have to spell out the class of functions (of the measure $\mu_t$) used to test the process on which this operator or kernel is acting. By construction, the functions we are going to consider are polynomials in the moments of the measure $\mu_t$ and their appropriate completions (that is, say, convergent series of the moments of the measure $\mu_t$, with a finite radius of convergence).

Let $\varphi_j$, $j = 1, \cdots, n$, be test functions and let $\mu_t[\varphi_j]$ be the corresponding moments. Let $f$ be a function of $n$ variables, $f(\vec{\mu}) = f(\mu_1, \cdots, \mu_n)$, defined by its series expansion (with a finite radius of convergence). The class of functions (of the measure $\mu_t$) we consider are defined by

$$F_f^{\vec{\varphi}}(\mu_t) := f(\mu_t[\varphi_1], \cdots, \mu_t[\varphi_n]).$$

We set $f^{\vec{\varphi}} = f \circ \vec{\varphi}$, that is $f^{\vec{\varphi}}(x) = f(\varphi_1(x), \cdots, \varphi_n(x))$.

We can then state:

**Proposition:** *Let $\mu_0$ be the initial condition $\mu_0 = \mu_{t=0}$.*

*At $\gamma \to \infty$ the limiting process for the measure $\mu_t$ is that of measures concentrated on a Brownian motion started at an initial position chosen randomly with distribution $\mu_0$. That is* [3]:

$$\lim_{\gamma \to \infty} d\mu_t = \delta_{Y_t}, \quad Y_t := B_{\frac{D}{2}t} \text{ with } Y_{t=0} \ \mu_0 -\text{distributed}, \tag{12}$$

*with $B_t$ a normalized Brownian motion. The limit is weak in the sense that it holds for all moments $F_f^{\vec{\varphi}}(\mu_t)$. Namely*

$$\lim_{\gamma \to \infty} \mathbb{E}\big[F_f^{\vec{\varphi}}(\mu_t)\big] = \mathbb{E}_{\mu_0}[f^{\vec{\varphi}}(Y_t)] = \mu_0[e^{t\frac{D}{2}\Delta} \cdot f^{\vec{\varphi}}], \tag{13}$$

*with $\mathbb{E}_{\mu_0}$ refers to the measure on Brownian motion with initial conditions distributed according to $\mu_0$.*

**Proof:** The proof goes in three steps: (i) first identify the differential operators (acting on $F_f^{\vec{\varphi}}$) generating the process; (ii) then identify the differential operator echoing the monitoring and look at its limiting action at large $\gamma$; and (ii) finally use a perturbation theory to deal with the effect of Lindblad dynamics on top of the monitoring.

(i) Let $F_f^{\vec{\varphi}}(\mu_t)$ be some moments of the random measure $\mu_t$. Using eq.(11) and Itô calculus, it is easy to check (as for general stochastic process generated by a stochastic differential equation) that these moments satisfies a stochastic differential equation of the form $dF_f^{\vec{\varphi}}(\mu_t) = (\mathcal{D} \cdot F_f^{\vec{\varphi}})(\mu_t) dt + \text{noise}$, with $\mathcal{D}$ a second order differential operator. Equivalently, their expectation reads

$$\mathbb{E}_{\mu_0}\big[F_f^{\vec{\varphi}}(\mu_t)\big] = \big([e^{t\mathcal{D}}] \cdot F_f^{\vec{\varphi}}\big)(\mu_0).$$

The form of eq.(11) ensures that the second order differential operator $\mathcal{D}$ decomposes as $\mathcal{D} = \mathcal{D}_0 + 4\gamma^2 \mathcal{D}_2$. Here $\mathcal{D}_0$ is a first order differential operator whose action on linear functions is such that $\mathcal{D}_0 \cdot \mu[\varphi] := \mu[\frac{D}{2}\Delta\varphi]$, by definition. It is extended to any functions $F_f^{\vec{\varphi}}(\mu)$ via Leibniz's rules:

$$\big(\mathcal{D}_0 \cdot F_f^{\vec{\varphi}}\big)(\mu) = \sum_{k=1}^n \mu[\frac{D}{2}\Delta\varphi_k](\nabla_k f)(\mu[\varphi_1], \cdots, \mu[\varphi_n]).$$

The operator $\mathcal{D}_2$ is a second order differential operator whose action on first and second moments is $\mathcal{D}_2 \cdot \mu[\varphi] = 0$ and $\mathcal{D}_2 \cdot \mu[\varphi_1]\mu[\varphi_2] = \mu[x \cdot \varphi_1]^c \mu[x \cdot \varphi_2]^c$, by definition. It is extended

---

[3]$\delta_y$ denotes de Dirac measure centered at $y$: $\delta_y := \delta(x - y)dx$.

to any functions $F_f^{\vec\varphi}(\mu)$ as a second order differential operator:

$$(\mathcal{D}_2 \cdot F_f^{\vec\varphi})(\mu) = \frac{1}{2} \sum_{k,l=1}^{n} \mu[x \cdot \varphi_k]^c \, \mu[x \cdot \varphi_l]^c \, (\nabla_k \nabla_l f)(\mu[\varphi_1], \cdots, \mu[\varphi_n]).$$

Both operators $\mathcal{D}_2$ and $\mathcal{D} = \mathcal{D}_0 + 4\gamma^2 \mathcal{D}_2$ are non-positive, because they are both associated to well defined stochastic differential equations.

(ii) The operator $4\gamma^2 \mathcal{D}_2$ is the one associated to QND $X$-measurement (without extra evolution). From the analysis of the previous, we know that in a pure QND monitoring of the position the measure $\mu_t$ behave at large $\gamma$ as

$$\mathsf{d}\mu_t(x)|^{\mathrm{qnd}} = \frac{1}{\mathcal{Z}_t} e^{-2\gamma^2 t \left( (x-\bar x)^2 + O(\gamma^{-1}) \right)} \, \mathsf{d}\mu_0(x),$$

with $\bar x$ $\mu_0$-distributed and $\mathcal{Z}_t \simeq \tilde\mu_0(\bar x) \sqrt{\pi/2\gamma^2 t}$ (with $\mathsf{d}\mu_0(x) = \tilde\mu_0(x)dx$). Thus we have

$$\lim_{\gamma\to\infty} \mathsf{d}\mu_t(x)|^{\mathrm{qnd}} = \delta_{\bar x},$$

with $\bar x$ $\mu_0$-distributed. Alternatively, this implies that $\mathbb{E}^{\mathrm{qnd}}\big[\mu_t[\varphi_1]\cdots\mu_t[\varphi_n]\big] \to \mu_0[\varphi_1\cdots\varphi_n]$, as $\gamma \to \infty$, which yields that

$$\lim_{\gamma\to\infty} \mathbb{E}^{\mathrm{qnd}}_{\mu_0}\big[F_f^{\vec\varphi}(\mu_t)\big] = \mu_0[f^{\vec\varphi}],$$

for any function $f$. Since $[e^{4\gamma^2 t \mathcal{D}_2}]$ is the transition kernel for QND measurements, this is equivalent to

$$\lim_{\gamma\to\infty} \Big([e^{4\gamma^2 t \mathcal{D}_2}] \cdot F_f^{\vec\varphi}\Big)(\mu_0) = \mu_0[f^{\vec\varphi}].$$

From this we learn that:

– The kernel of $\mathcal{D}_2$ is made of linear functions: $\mathrm{Ker}\mathcal{D}_2 = \{\mu[\varphi], \ \varphi \text{ test function}\}$.

– Let $\Pi$ be the projector on $\mathrm{Ker}\mathcal{D}_2$ defined by $\Pi := \lim_{a\to\infty} e^{a\mathcal{D}_2}$. Let $F_f^{\vec\varphi}|_\parallel$ be the projection of $F_f^{\vec\varphi}$ on $\mathrm{Ker}\mathcal{D}_2$, defined by $F_f^{\vec\varphi}|_\parallel = \Pi \cdot F_f^{\vec\varphi}$. Then $F_f^{\vec\varphi}|_\parallel(\mu) = \mu[f^{\vec\varphi}]$.

(iii) We now study the (original) process whose generator is $\mathcal{D} = \mathcal{D}_0 + 4\gamma^2 \mathcal{D}_2$. By construction, we have

$$\mathbb{E}_{\mu_0}\big[F_f^{\vec\varphi}(\mu_t)\big] = \Big([e^{t(\mathcal{D}_0 + 4\gamma^2 \mathcal{D}_2)}] \cdot F_f^{\vec\varphi}\Big)(\mu_0).$$

It is clear that if $F_f^{\vec\varphi}|_\parallel = 0$ (that is $F_f^{\vec\varphi}|$ has no linear component), then $\lim_{\gamma\to\infty} \mathbb{E}_{\mu_0}\big[F_f^{\vec\varphi}(\mu_t)\big] = 0$. That is: only expectations of functions in $\mathrm{Ker}\mathcal{D}_2$ have a non trivial limit as $\gamma \to \infty$, and the limiting process is reduced to that space of zero modes (i.e. to that kernel).

We now use an algebraic strategy to prove convergence. Since $\mathcal{D}_0 : \mathrm{Ker}\mathcal{D}_2 \to \mathrm{Ker}\mathcal{D}_2$, perturbation theory tells us (as recalled below) that the dynamics on $\mathrm{Ker}\mathcal{D}_2$ reduces to

$$\lim_{\gamma\to\infty} \mathbb{E}_{\mu_0}\big[F_f^{\vec\varphi}(\mu_t)\big] = \Big([e^{t\mathcal{D}_0} \circ \Pi] \cdot F_f^{\vec\varphi}\Big)(\mu_0),$$

or equivalently,

$$\lim_{\gamma\to\infty} \mathbb{E}_{\mu_0}\big[F_f^{\vec\varphi}(\mu_t)\big] = [e^{t\mathcal{D}_0}] \cdot \mu_0[f^{\vec\varphi}]. \tag{14}$$

This last equation is equivalent to the claim (13) because $[e^{t\mathcal{D}_0}] \cdot \mu_0[f^{\vec\varphi}] = \mu_0[e^{t\frac{D}{2}\Delta} \cdot f^{\vec\varphi}]$.

It is thus remains to argue for eq.(14). Let $K_t^\gamma := [e^{t(\mathcal{D}_0 + 4\gamma^2 \mathcal{D}_2)}]$ and $\Pi_t^\gamma := [e^{t 4\gamma^2 \mathcal{D}_2}]$. We have $\partial_t K_t^\gamma = K_t^\gamma(\mathcal{D}_0 + 4\gamma^2 \mathcal{D}_2)$ and $\lim_{\gamma\to\infty} \Pi_t^\gamma = \Pi$ with $\Pi$ the project on linear function (i.e. on $\mathrm{Ker}\mathcal{D}_2$). Let $F \in \mathrm{Ker}\mathcal{D}_2$. Then $\partial_t(K_t^\gamma \cdot F) = K_t^\gamma \mathcal{D}_0 \cdot F$ (because $\mathcal{D}_2 \cdot F = 0$). Equivalently

$\partial_t(K_t^\gamma \Pi) = K_t^\gamma(\mathcal{D}_0 \Pi)$. Now because $\mathcal{D}_0$ maps $\mathrm{Ker}\mathcal{D}_2$ onto $\mathrm{Ker}\mathcal{D}_2$ we have $\mathcal{D}_0 \Pi = \Pi \mathcal{D}_0 \Pi$, and thus $\partial_t(K_t^\gamma \Pi) = (K_t^\gamma \Pi)(\Pi \mathcal{D}_0 \Pi)$. Integrating and taking the large $\gamma$ limit yields eq.(14). $\qquad\square$

Let us end this sub-section by a remark. It is easy to verify that the dynamical equation (10) admits a separation of variables so that its general solutions are density kernels $\rho_t(x, y)$ of the following form

$$\rho_t(x, y) = \sigma_0\left(\frac{x-y}{2}\right) \cdot e^{-\frac{\gamma^2}{2}(x-y)^2 t} \cdot \tilde{\mu}_t\left(\frac{x+y}{2}\right),$$

with $\sigma_0$ an arbitrary function (normalized to $\sigma_0(0) = 1$) and $\tilde{\mu}_t$ the density (with respect o the Lebesgue measure) of the measure $d\mu_t$, i.e. $d\mu_t(x) = \tilde{\mu}_t(x)dx$. This gives the complete solution of the position monitoring a simple quantum diffusion. It is clear that, except for the observable position (the observable $X$), the measures associated to any other system observables have no well defined large $\gamma$ limit. This in particular holds true for the momentum observable $P$, as expected.

## 3.2 Generalization : General stochastic diffusion

We can reverse the logic and ask ourselves whether it is possible to obtain any stochastic differential equation, of the form $dY_t = U(Y_t)dt + V(Y_t)dB_t$, as the strong monitoring limit of a quantum dynamical systems. That is: we ask whether, given two real functions $U(y)$ and $V(y)$ (sufficiently well-behaved), we can choose a Lindbladian $L$ such that the large $\gamma$ limit of the quantum trajectories

$$d\rho_t = L(\rho_t)dt - \frac{\gamma^2}{2}[X, [X, \rho_t]]dt + \gamma\big(X\rho_t + \rho_t X - 2\langle X\rangle_t \rho_t\big)dW_t, \tag{15}$$

leads to solutions of the stochastic differential equation $dY_t = U(Y_t)dt + V(Y_t)dB_t$.

**Proposition:** *Let $L = L_U + L_V$ be the sum of two Lindblad operators such that their duals $L_U^*$ and $L_V^*$ act on any observable $\hat{\varphi}$ on $\mathcal{H} = L^2(\mathbb{R})$, as follows (recall that $V(X)^* = V(X)$ and $U(X)^* = U(X)$)*

$$L_V^*(\hat{\varphi}) = V(X)P\hat{\varphi}PV(X) - \frac{1}{2}\big(\hat{\varphi}V(X)P^2V(X) + V(X)P^2V(X)\hat{\varphi}\big),$$

$$L_U^*(\hat{\varphi}) = \frac{i}{2}[U(X)P + PU(X), \hat{\varphi}],$$

*with $X$ the position observable and $P$ the momentum observable (such that $[X, P] = i$).*
*Let $\mu_t$ be the measure on the real line induced by $\rho_t$ via $\int d\mu_t(x)\varphi(x) = \mathrm{Tr}\big(\rho_t \varphi(X)\big)$ for any function $\varphi$ with $\rho_t$ solution of eq.(15).*
*Then, in the large $\gamma$ limit, $\mu_t$ concentrates on solutions of the stochastic differential equation $dY_t = U(Y_t)dt + V(Y_t)dB_t$, in the sense that*

$$\lim_{\gamma \to \infty} d\mu_t = \delta_{Y_t} \quad \text{with } dY_t = U(Y_t)dt + V(Y_t)dB_t.$$

**Proof:** Recall the definition $\mu_t[\varphi] := \int d\mu_t(x)\varphi(x) = \mathrm{Tr}\big(\rho_t \varphi(X)\big)$. By duality, if $\rho_t$ evolves according to eq.(15), then

$$d\mu_t[\varphi] = \mu_t[\hat{D} \cdot \varphi]dt + 2\gamma \mu_t[x \cdot \varphi]^c dW_t,$$

with $\mu_t[x \cdot \varphi]^c = \mu_t[x \cdot \varphi] - \mu_t[x]\mu_t[\varphi]$ as before and $\hat{D}$ a linear operator on function $\varphi$ such that

$$\mu_t[\hat{D} \cdot \varphi] = \mathrm{Tr}\big(L(\rho_t)\varphi(X)\big) = \mathrm{Tr}\big(\rho_t L^*(\varphi(X))\big),$$

because $\mathrm{Tr}\big(L(\rho_t)\,\varphi(X)\big) = \mathrm{Tr}\big(\rho_t\,L^*(\varphi(X))\big)$ by definition the dual Lindbladian $L^*$. The operator $\hat{D}$ exists and is well-defined because, as we shall see, our choice of $L$ ensures that $L^*(\varphi(X))$ is again a function of the observable $X$. To prove the claim we need to check that $L^*$ is such that

$$L^*(\varphi(X)) = \frac{1}{2} V^2(X) \partial_x^2 \varphi(X) + U(X) \partial_x \varphi(X) =: (D_{\mathrm{st}} \cdot \varphi)(X),$$

because the differential operator associated to the SDE $dY_t = U(Y_t) dt + V(Y_t) dB_t$ is $D_{\mathrm{st}} = \frac{1}{2} V^2(x) \partial_x^2 + U(x) \partial_x$. Now, if $\hat{\varphi} = \varphi(X)$, so that it commutes with $V(X)$ and $U(X)$, we have

$$
\begin{aligned}
L_V^*(\varphi(X)) &= -\frac{1}{2} V(X) [P, [P, \varphi(X)]] V(X) = +\frac{1}{2} V(X)^2 \partial_x^2 \varphi(X), \\
L_U^*(\varphi(X)) &= \frac{i}{2} \big( U(X)[P, \varphi(X)] + [P, \varphi(X)] U(X) \big) = U(X) \partial_x \varphi(X),
\end{aligned}
$$

so that $(L_V^* + L_U^*)\varphi(X) = (D_{\mathrm{st}} \cdot \varphi)(X)$ as required.

The rest of the proof is as before. We look at the functions $F_f^{\vec{\varphi}}(\mu_t)$. By identical arguments (with $D_{\mathrm{st}}$ replacing the Laplacian $\Delta$), we then have

$$\lim_{\gamma \to \infty} \mathbb{E}_{\mu_0}\big[F_f^{\vec{\varphi}}(\mu_t)\big] = \mu_0[e^{t D_{\mathrm{st}}} \cdot f^{\vec{\varphi}}]. \tag{16}$$

In parallel, let $d\mu_t^\infty := \delta_{Y_t}$ with $Y_t$ solution of $dY_t = U(Y_t) dt + V(Y_t) dB_t$ with initial condition $Y_{t=0}$ $\mu_0$-distributed. Then, $\mu_t^\infty[\varphi] = \varphi(Y_t)$ and $F_f^{\vec{\varphi}}(\mu_t^\infty) = f^{\vec{\varphi}}(Y_t)$, and we have

$$\mathbb{E}\big[F_f^{\vec{\varphi}}(\mu_t^\infty)\big] = \mathbb{E}_{\mu_0}[f^{\vec{\varphi}}(X_t)] = \mu_0[e^{t D_{\mathrm{st}}} \cdot f^{\vec{\varphi}}]. \tag{17}$$

Comparing eq.(16) and eq.(17) proves the claim. $\qquad\square$

Let us end this sub-section by a remark. This construction generalizes to higher dimensional systems. Indeed considered a system of stochastic differential equations, $dY_t^j = U^j(Y_t) dt + V_a^j(Y_t) dB_t^a$, (with implicit summation over repeated indices) on $M$ variables $Y^j$ driven by $N$ motions $B_t^a$ with quadratic variations $dB_t^a dB_t^d = \kappa^{ab} dt$. We may then ask under which conditions a quantum system concentrates along trajectories solutions of these SDEs. Of course, the system has to be in dimension $M$ with Hilbert space $\mathcal{H} = L^2(\mathbb{R}^M)$. Let us consider the evolution equation (15) generalized in dimension $M$ (with monitoring of the $M$ observables $X^j$) with Lindblad operator $L = L_U + L_V$ with (with implicit summation on repeated indices)

$$
\begin{aligned}
L_V^*(\hat{\varphi}) &= \kappa^{ab} \big( V_a^j(X) P_j \,\hat{\varphi}\, P_k V_b^k(X) - \frac{1}{2} \big( \hat{\varphi}\, V_a^j(X) P_j P_k V_b^k(X) + V_a^j(X) P_j P_k V_b^k(X) \,\hat{\varphi} \big), \\
L_U^*(\hat{\varphi}) &= \frac{i}{2} [U^j(X) P_j + P_j U^j(X), \hat{\varphi}],
\end{aligned}
$$

with $P_j$ the momentum operator conjugated to the position observable $X^j$ (i.e. $[X^j, P_k] = i\delta_k^j$). It is then easy to check that the measure on $\mathbb{R}^M$ associated to $X^j$ and induced by the density matrix evolving according to the $M$-dimensional generalization of eq.(15) concentrates in the large $\gamma$ limit along the trajectories solutions of $dY_t^j = U^j(Y_t) dt + V_a^j(Y_t) dB_t^a$.

It remains an open question to decipher what are the stochastic processes describing the strong monitoring limit for Lindladians not quadratic in the impulsion operators.

# 4 Monitoring continuous spectrum observable with Hamiltonian dynamics

The aim of this section is to analyze similarly the large monitoring limit for a system undergoing a Hamiltonian non dissipative dynamics. We consider a particle on the real line with Hilbert

space $\mathcal{H} = L^2(\mathbb{R})$ and monitoring its positions. The density matrix dynamical equation is (we put back the Planck constant for later convenience)

$$d\rho_t = -\frac{i}{\hbar}[H, \rho_t]\,dt - \frac{\gamma^2}{2}[X, [X, \rho_t]]\,dt + \gamma\big(X\rho_t + \rho_t X - 2\langle X\rangle_t \rho_t\big)\,dW_t, \qquad (18)$$

for some Hamiltonian $H$. As is well known, this equation preserves pure states (by construction, because monitoring preserves the state purity), so that we can equivalently write it on wave functions $\psi_t(x)$ [18–21]:

$$d\psi_t(x) = -\frac{i}{\hbar}(H\psi_t)(x)\,dt - \frac{\gamma^2}{2}(x - \langle X\rangle_t)^2\,\psi_t(x)\,dt + \gamma\,(x - \langle X\rangle_t)\,\psi_t(x)\,dW_t, \qquad (19)$$

with $\langle X\rangle_t = \int dx\, x|\psi_t(x)|^2$ and $(H\psi_t)(x) = -\frac{\hbar^2}{2m}\partial_x^2\psi_t(x) + V(x)\psi_t(x)$ for a (non-relativistic) particle of mass $m$ in a potential $V$.

As recalled in the introduction, equation (19) encodes for three different regimes [23–28, 30]: (a) a collapse regime, (b) a classical regime, and c) a diffusive regime. The aim of this section is to show that we may define a scaling limit which describes the cross-over from the classical regime to the diffusive regime.

## 4.1 A simple case: Particle in a harmonic potential

Let us start with this simple case which includes a free particle. It will allow us to decipher what strong monitoring limit we may expect and which features may be valid in a more general setting. We closely follow methods of analysis used refs. [28, 30, 32, 33].

Let $V(x) = \frac{1}{2}m\Omega^2 x^2$ be the potential. As is well known, eq.(19) is better solved by representing the wave function as $\psi_t(x) = \phi_t(x)/\sqrt{Z_t}$ with the normalization $Z_t$ and $\phi_t(x)$ solution of the linear equation

$$d\phi_t(x) = i\hbar^{-1}\Big(\frac{\hbar^2}{2m}\partial_x^2\phi_t(x) - V(x)\phi_t(x)\Big)\,dt - \frac{\gamma^2}{2}x^2\,\phi_t(x)\,dt + \gamma\,x\,\phi_t(x)\,dS_t, \qquad (20)$$

where $S_t$ is the monitoring signal (with $dS_t^2 = dt$), solution of $dS_t = 2\gamma\langle X\rangle_t\,dt + dW_t$. The normalization factor $Z_t$ is such that $dZ_t = 2\gamma\,\langle X\rangle_t\,Z_t\,dS_t = 2\gamma\big(\int dx\,x|\phi_t(x)|^2\big)\,dS_t$. Besides the frequency $\Omega$ associated to the harmonic potential, there is another frequency scale $\omega$ and a length scale $\ell$, both arising from the position monitoring, with

$$\ell^4 := \frac{\hbar}{m\gamma^2}\,, \quad \omega^2 := \frac{\hbar\gamma^2}{m}.$$

Eq.(20) is a Schrödinger equation in a complex harmonic potential and can be exactly solved via superposition of Gaussian wave packets. Thus, as in [30] we take a Gaussian ansatz for the un-normalized wave function written as

$$\phi_t(x) = \phi_0 \exp\big(-a_t(x - \bar{x}_t)^2 + i\bar{k}_t x + \alpha_t\big), \qquad (21)$$

where all the time-indexed quantities have to be thought as stochastic variables. For a single Gaussian packet ansatz – the case we shall consider –, $\bar{x}_t$ and $\bar{k}_t$ are the mean position and mean wave vector. This single Gaussian packet is then solution of eq.(19) if [26–28, 30]

$$da_t = \ell^{-2}\Big(1 - i2\ell^4 a_t^2 + i\frac{\Omega^2}{2\omega^2}\Big)\omega\,dt\,,$$

$$d\bar{x}_t = \bar{v}_t\,dt + \frac{\sqrt{\omega}}{2\ell a_t^R}\,dW_t\,,$$

$$d\bar{v}_t = -\Omega^2\bar{x}_t\,dt - \ell\omega^{\frac{3}{2}}\frac{a_t^I}{a_t^R}\,dW_t\,,$$

with $\bar{v}_t$ is the mean velocity. Here $a_t^R (a_t^I)$ denotes the real (imaginary) part of $a_t$ respectively (i.e. $a_t = a_t^R + i a_t^I$).

From these equations, it is clear that $\tau_c = 1/\omega$ is the typical time for the wave function to collapse. After a typical time of order $\tau_c$, the Gaussian packet reaches its minimum size with $a_t \simeq a_\infty$ for $t \gg \tau_c$ with

$$a_\infty = \Big(\frac{1}{2i\ell^4}\big(1 + i\frac{\Omega^2}{\omega^2}\big)\Big)^{1/2}.$$

Taking $\omega \to \infty$ while keeping $\Omega$ fixed allows us to simplify,

$$a_\infty = \frac{e^{-i\pi/4}}{\sqrt{2}}\,\ell^{-2}. \tag{22}$$

In other words, monitoring stabilizes the wave function in a Gaussian wave packet with constant (minimal) width $\ell$. In this collapsed wave packet, the position and velocity dispersions are

$$\sigma_x = 2^{-1/4}\ell, \quad \sigma_v = 2^{-3/4}\omega\ell.$$

After this transient collapsing period, for $t \gg \tau_c$, the mean position and velocity evolve according to

$$d\bar{x}_t = \bar{v}_t\,dt + \sqrt{2\omega\ell^2}\,dW_t, \tag{23}$$

$$d\bar{v}_t = -\Omega^2\bar{x}_t\,dt + \ell\omega^{\frac{3}{2}}\,dW_t. \tag{24}$$

We now may wonder if there is a well defined strong monitoring limit (i.e. a limit $\gamma \to \infty$). On physical ground, this limit should be such that the time to collapse vanishes, that is $\tau_c \to 0$ or equivalently $\omega \to \infty$. It is then clear from eqs.(23,24) above that $\ell$ should simultaneously vanish for this limit to make sense, so that the strong monitoring limit is the double scaling limit $\omega \to \infty$, $\ell \to 0$. A closer inspection of eqs.(23,24) shows that we should take this double limit with $\varepsilon := \omega^3\ell^2$ fixed (so that $\sqrt{\ell^2\omega} \to 0$). Note that, in this limit, the wave packet is localized both in space and in velocity, $\sigma_x \to 0$ and $\sigma_v \to 0$, so it is actually a classical limit (i.e. $\gamma \to \infty$ and $\hbar \to 0$ with $\hbar\gamma/m$ fixed).

We can summarize this discussion:

**Proposition:** *In the double limit $\omega \to \infty$ and $\ell \to 0$ at $\varepsilon := \omega^3\ell^2 = \hbar^2\gamma^2/m^2$ fixed, solution of the quantum trajectory equation (19) in an harmonic potential $V(x) = \frac{1}{2}m\Omega x^2$ localizes in the sense that the probability density $|\psi_t(x)|^2 = \delta_{\bar{x}_t}$ with $\bar{x}_t$ solution of the stochastic equations*

$$d\bar{x}_t = \bar{v}_t\,dt, \tag{25}$$

$$d\bar{v}_t = -\Omega^2\bar{x}_t\,dt + \sqrt{\varepsilon}\,dW_t. \tag{26}$$

This behavior describes the cross-over from a semi-classical behavior, which occurs just after the transient collapsing period, to the diffusion behavior due to monitoring back action. As is well known, eqs.(25, 26) can be solved exactly with solution:

$$\bar{x}_t = x_0\cos(\Omega t) + \sqrt{\frac{\varepsilon}{\Omega^2}}\int_0^t dW_s\sin(\Omega(t-s)),$$

where we chose for simplicity the initial conditions $x(t=0) = x_0$, $v(t=0) = 0$. It reflects the cross-over behavior from the classical solution $\bar{x}_t \simeq x_0\cos(\Omega t)$ at small time to the diffusion behavior $\bar{x}_t \simeq \sqrt{\frac{\varepsilon}{\Omega^2}}\int_0^t dW_s\sin(\Omega(t-s))$ at large time. The fuzziness of the trajectory can be testified by computing the variance of the position. We have $(\Delta\bar{x}_t)^2 = \frac{\varepsilon}{\Omega^2}(\frac{2\Omega t - \sin(2\Omega t)}{4\Omega})$, so that $(\Delta\bar{x}_t)^2 \simeq \varepsilon t/2\Omega^2$ for $\Omega t \gg 1$ which is typical of a diffusive behavior and $(\Delta\bar{x}_t)^2 \simeq \frac{\varepsilon t^3}{3}$ for $\Omega t \ll 1$ which can be interpreted as a state localized with accuracy $\simeq t^3$ for small times. The two behaviors are showed in fig.1.

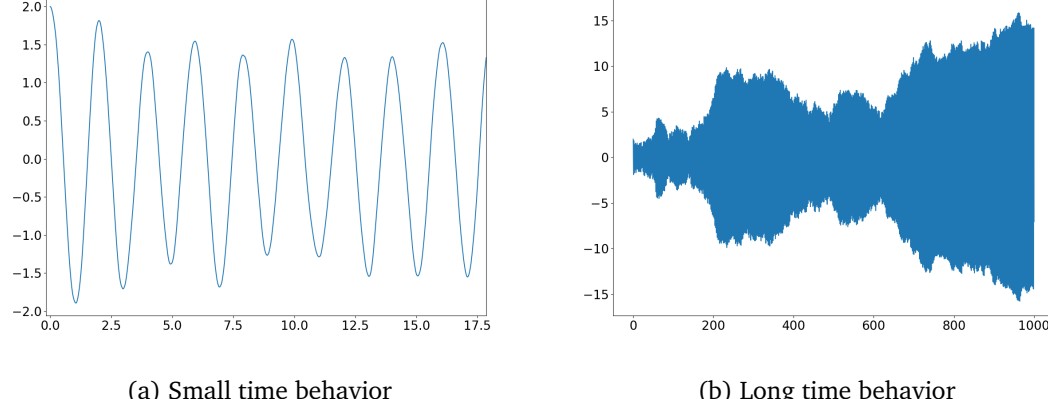

(a) Small time behavior  (b) Long time behavior

Figure 1: Typical behaviors of a particle trapped in an harmonic oscillator in the strong monitoring regime. The plots correspond to the evolution of the position of the particle viewed at different time scales and evolving according to (25,26) with $\Omega = 1$, $\epsilon = 1$, $x_0 = 2$, $v_0 = 0$. We clearly have a transition from an oscillatory behavior at time of order $1/\Omega$ to a diffusive regime for $t \gg 1/\Omega$.

## 4.2 Generalization: A particle in a smooth potential

We can now borrow the previous observation to state what the strong monitoring limit could be for a particle in an arbitrary potential. As suggested sometime ago [26–28, 31], after a transient time of order $\tau_c = 1/\omega$ with $\omega^2 := \hbar\gamma^2/m$, continuous monitoring of the position leads to a collapse of the wave function on a Gaussian state with a minimal width of order $\ell$ with $\ell^4 := \hbar/m\gamma^2$. In view of the previous analysis we are lead to suggest that

**Conjecture:** *In the double limit $\gamma \to \infty$ and $\hbar/m \to 0$ at $\varepsilon := \hbar^2\gamma^2/m^2$ fixed, solution of the quantum trajectory equation (19) in a potential $V(x)$ localizes at a position $\bar{x}_t$ solution of the Langevin equation*

$$d\bar{x}_t = \bar{v}_t\, dt\, , \tag{27}$$

$$d\bar{v}_t = -\frac{1}{m}\partial_x V(\bar{x}_t)\, dt + \sqrt{\varepsilon}\, dW_t\, . \tag{28}$$

Let us give a few arguments in favour of this claim. After the transient collapse time, we can take the Gaussian wave packet (21) for the (unnormalized) wave function. Taylor expanding the potential around the mean position $\bar{x}_t$ and keeping only the terms up to second order leads to the stochastic equations for the width, the mean position and the mean velocity :

$$da_t = \left(\gamma^2 - 2i\frac{\hbar}{m}a_t^2 + \frac{i}{2\hbar}\partial_x^2 V(\bar{x}_t)\right)dt\, ,$$

$$d\bar{x}_t = \bar{v}_t\, dt + \frac{\gamma}{2a_t^R}\, dW_t\, ,$$

$$d\bar{v}_t = -\frac{1}{m}\partial_x V(\bar{x}_t)\, dt - \frac{\gamma\hbar a_t^I}{ma_t^R}\, dW_t\, ,$$

with $a_t = a_t^R + ia_t^I$ and where we supposed that

$$(x - \bar{x}_t)^3\partial_x^3 V(\bar{x}_t) \ll (x - \bar{x}_t)^2\partial_x^2 V(\bar{x}_t), \tag{29}$$

for every $x$, $\bar{x}_t$. As for the harmonic potential, at time $t \gg \tau_c$ the width reaches its stationary values $a_\infty = \left(\frac{1}{2i\ell^4}(1 + \frac{i}{2\hbar\gamma^2}\partial_x^2 V(\bar{x}_t))\right)^{1/2}$. Demanding that this width be independent of dynamical aspects, that is, independent of the position, imposes $|\partial_x^2 V(\bar{x}_t)| \ll \hbar\gamma^2$. This is indeed

satisfied in the double scaling limit (if the potential is smooth enough) and it coincides with the condition of ref. [32, 33] forcing a localisation of the wave packet. We then simply have $a_\infty = \frac{e^{-i\frac{\pi}{4}}}{\sqrt{2}}\ell^{-2}$ and, as in the harmonic case, the position and speed dispersions are given by $\sigma_x = 2^{-1/4}\ell$, $\sigma_v \equiv 2^{-3/4}\omega\ell$, in this limit.

Plugging these asymptotic values in the above equations and taking the scaling limit at $\varepsilon := \hbar^2\gamma^2/m^2$ fixed, leads to the Langevin equations (27,28).

Remark that we can make an a posteriori self-consistent check for the approximation (29). The typical length on which the wave function admits non zero value in the $\omega \to \infty$, $\ell \to 0$ limit at $\varepsilon$ fixed scales like $\ell$ so that condition (29) amounts to $\ell\, \partial_x^3 V(\bar{x}_t) \ll \partial_x^2 V(\bar{x}_t)$. This is indeed valid for $\ell \to 0$ if the potential is smooth enough (with finite second and third spatial derivative) everywhere in space.

## Acknowledgements

This work was in part supported by the ANR project "StoQ", contract number ANR-14-CE25-0003. D.B. thanks C. Pellegrini for discussions and for his interest in this work.

## A  Exchangeable processes and QND measurements

We here give the details of the arguments needed to prove the main proposition of the Section 2. Before discussing the continuous time limit, it is useful to have a look at the case of discrete time, which is more elementary. Our point is to show two equivalent descriptions of the monitoring during a non-demolition measurement process.

### A.1  Discrete time monitoring

The evolution equation for the diagonal elements of the density matrix (this is a measure in general, a subtlety that becomes unavoidable for an observable with continuous spectrum) in the pointer states basis in repeated quantum non-demolition measurements in discrete time reads,

$$\mathrm{d}\mu_{n+1}(\alpha) = \frac{\mathrm{d}\mu_n(\alpha)p(i|\alpha)}{\int \mathrm{d}\mu_n(\beta)p(i|\beta)} \text{ with probability } \int \mathrm{d}\mu_n(\beta)p(i|\beta),$$

where Greek letters $\alpha, \beta, \dots$ index pointer states and Latin letters $i, j, \cdots$ are the outcomes of the indirect measurements. The observation process is defined by

$$S_n(i) := \#\{m \le n, m^{th} \text{ measurement has given outcome } i\} =: \sum_{m=1}^{n} \varepsilon_m(i),$$

so that $\varepsilon_n(i) = 1$ if the outcome of the $n^{th}$ measurement is $i$ (probability $\int \mathrm{d}\mu_{n-1}(\beta)p(i|\beta)$) and 0 otherwise.

The natural situation of the experimenter is to have only access to the observation process, and possibly to the initial condition. From this viewpoint, the fate of the observation process $S_n$ and of $\mathrm{d}\mu_n$ at large $n$ is not so easy to decipher.

However the evolution equations are easily "solved" to yield the joint law of $\varepsilon_1, \cdots, \varepsilon_n$ which is

$$\mathrm{Prob}(\varepsilon_1, \cdots, \varepsilon_n) = \int \mathrm{d}\mu_0(\alpha) \prod_i p(i|\alpha)^{S_n(i)},$$

and the value of $d\mu_n$ which is

$$d\mu_n(\alpha) = d\mu_0(\alpha) \frac{\prod_i p(i|\alpha)^{S_n(i)}}{\int d\mu_0(\beta) \prod_i p(i|\beta)^{S_n(i)}}.$$

This law, which involves only the random variables $S_n = \sum_{m=1}^n \varepsilon_m$ is obviously invariant under permutation of $\varepsilon_1, \cdots, \varepsilon_n$, which expresses the property that the sequence $\varepsilon_1, \cdots, \varepsilon_n$ is exchangeable. De Finetti's theorem expresses that if this holds for each $n = 1, 2, \cdots$ then the random variables $\varepsilon_1, \varepsilon_2, \cdots$ are conditionally independent and identically distributed. This is also apparent from the explicit formula for the law: conditionally on a choice of $\alpha$ (sampled with the law $d\mu_0(\alpha)$) the joint law of $\varepsilon_1, \cdots, \varepsilon_n$ is

$$\prod_i p(i|\alpha)^{S_n(i)} = \prod_{m=1}^n \prod_{i_m} p(i_m|\alpha)^{\varepsilon_m(i_m)},$$

which says that the $m^{th}$ measurement yields observation of $i_m$ with probability $p(i_m|\alpha)$ independently of the other observations. From this viewpoint, it is clear that the sequence of frequencies $\frac{S_n(i)}{n}$ converges almost surely to $p(i|\alpha)$ so that if the conditional probability distributions $p(\cdot|\beta)$ are distinct for different $\beta$s the asymptotics of the observation process allow to recover the value $\alpha$ sampled initially.

To summarize, the natural situation of an experimenter if to have access at time $n$ to $\varepsilon_1, \cdots, \varepsilon_n$, but the law of the observation process is exactly the same as if, before the experiment begins, a "cheater" samples the state of the system with $d\mu_0$ so that he has access not only to $\varepsilon_1, \cdots, \varepsilon_n$ but also to $\alpha$. The cheater knows in advance the asymptotics (i.e. $\alpha$) while the experimenter discovers it only progressively as time goes by. To make the point again: the *same* process has two different interpretations depending on the information you have at your disposal.

De Finetti's theorem is also closely related to the notion of reverse martingales. Fix $0 < l < m$. Due to exchangeability, it is easy to see that knowing $\frac{S_n}{n} = \left(\frac{S_n(i)}{n}\right)_i$ for $n \geq m$, the best estimate (i.e. the conditional expectation) for $\frac{S_l}{l}$ is $\frac{S_m}{m}$. In words, the best estimate for a value in the past knowing the future is the present. This is the notion of reverse martingales, to be contrasted with (usual) martingales for whom the best estimate for a value in the future knowing the past is the present. It turns out that the notion of reverse martingales is even more rigid than that of martingales: without any conditions (apart from the existence of conditional expectations implied by their very definition), reverse martingales converge at large times, almost surely and in $\mathbb{L}^1$. Of course, in the case at hand, we can rely on the explicit formula for the law of $S_n$ and the strong law of large numbers to be sure that $\frac{S_n}{n}$ has a limit at large $n$ but this is deceptive: if $K_n$, $n \geq 0$, is the sequence of partial sums of independent identically distributed integrable random variables, then $\frac{K_n}{n}$ is an example of a reverse martingale, so the reverse martingale convergence theorem immediately implies the strong law of large numbers and yields a conceptual proof of it (see e.g. page 484 in [34]).

## A.2   Continuous time monitoring

Not only do the frequencies $\frac{S_n(i)}{n}$ converge at large $n$. In fact, a stronger property, a central limit theorem holds: if the limiting pointer state is $\alpha$, $\frac{S_n(i) - np(i|\alpha)}{n^{1/2}}$ converges to a Gaussian. Note that $\sum_i S_n(i) = n$ so there is one degree of freedom less than the number of possible measurement outcomes. To take a continuous time limit, the situation resembles that of random walks: one has to replace $p(i|\alpha)$ by $p_\delta(i|\alpha)$ where $\delta \searrow 0$ is the time increment, with $p_\delta(i|\alpha) = p_0(i) + O(\delta^{1/2})$ so that each observation has only a small effect (on the correct order of magnitude) on $d\mu$. Assuming for simplicity that the pointer state basis is indexed by a

real number $A$ and that $i$ takes only two values (so that there is only one degree of freedom) one is naturally led to an observation process described up to normalization by $S_t = B_t + At$, $t \in [0, +\infty[$ where $B_t$ is a standard Brownian motion and $A$ is sampled from an initial distribution $d\mu_0$. This is the description from the perspective of the "cheater", whose knowledge at time $t$ is $\alpha$ and $S_u, u \in [0, t]$, or $\alpha$ and $B_u, u \in [0, t]$. In more mathematical terms, the "cheater" observes the process via the filtration $\mathcal{G}_t := \sigma\{A \text{ and } S_u, u \in [0, t]\} = \sigma\{A \text{ and } B_u, u \in [0, t]\}$. Let us note that a general theorem (see e.g. page 322 in [35]) based on a natural extension of the notion of exchangeability ensures that $CB_t + At$ with $C, A$ random and $B_t$ $t \in [0, +\infty[$ an independent Brownian motion is the most general continuous exchangeable process on $[0, +\infty[$. A random conditional variance such as $C$ in the above formula plays no role in the forthcoming discussion.

Our goal is to get the description of the same process for the experimenter, who knows only $S_u, u \in [0, t]$ at time $t$, i.e. uses the filtration $\mathcal{H}_t := \sigma\{S_u, u \in [0, t]\}$. It is also useful to introduce the filtration $\mathcal{F}_t := \sigma\{B_u, u \in [0, t]\}$. The relations between the filtrations $\mathcal{F}_t, \mathcal{G}_t, \mathcal{H}_t$ are the clue to solve our problem. It is crucial that $\mathcal{F}_t$ is independent of $A$ and that $\mathcal{F}_t, \mathcal{H}_t \subset \mathcal{G}_t$, and we use these properties freely in conditional expectations in what follows. We let $\mathbb{E}$ denote the global (i.e. over both $A$ and the Brownian motion) expectation symbol.

## A.3  Change of filtration

The crucial computation is an identity for the joint law of the random variable $A$ and the process $S_t$.

**Proposition:** *Let $f$ be a nice (measurable and such that the following expectations make sense, non-negative or bounded would certainly do) function from $\mathbb{R}^{k+1}$ to $\mathbb{R}$, and $0 = t_0 < t_1 \cdots < t_k = t$. Then*

$$\mathbb{E}\left[f(A, S_{t_1}, \cdots, S_{t_k})\right] = \mathbb{E}\left[f(A, B_{t_1}, \cdots, B_{t_k})e^{AB_t - A^2 t/2}\right].$$

The general tool to understand such a formula is Girsanov's theorem, but in the case at hand, an easy (if tedious) explicit computation does the job. The idea is to write

$$
\begin{aligned}
\mathbb{E}\left[f(A, S_{t_1}, \cdots, S_{t_k})\right] &= \mathbb{E}\left[f(A, B_{t_1} + At_1, \cdots, B_{t_k} + At_k)\right] \\
&= \int d\mu_0(\alpha)\mathbb{E}\left[f(\alpha, B_{t_1} + \alpha t_1, \cdots, B_{t_k} + \alpha t_k)\right],
\end{aligned}
$$

where the first equality is the definition the observation process and the second makes use of the fact that the Brownian motion is independent of $A$. Thus we are left to prove the identity

$$\mathbb{E}\left[f(\alpha, B_{t_1} + \alpha t_1, \cdots, B_{t_k} + \alpha t_k)\right] = \mathbb{E}\left[f(\alpha, B_{t_1}, \cdots, B_{t_k})e^{\alpha B_t - \alpha^2 t/2}\right]$$

for every $\alpha \in \mathbb{R}$. This is done by writing the left-hand side using the explicit expression of the finite dimensional distributions of Brownian motion in terms of the Gaussian kernel and translating the integration variables $x_l$ associated to the positions at time $t_l$, $l = 1, \cdots, k$, by $\alpha t_l$. Setting $x_l + \alpha t_l = y_l$ (with the convention $x_0 = y_0 = 0$) one gets

$$-\frac{(x_l - x_{l-1})^2}{2(t_l - t_{l-1})} = -\frac{(y_l - y_{l-1})^2}{2(t_l - t_{l-1})} + \alpha(y_l - y_{l-1}) - \alpha^2(t_l - t_{l-1})/2,$$

which leads to a telescopic sum $\sum_{l=1}^k \alpha(y_l - y_{l-1}) - \alpha^2(t_l - t_{l-1})/2 = \alpha y_k - \alpha^2 t_k/2$ yielding an expression which is recognized as the right-hand side.

We use this identity to understand the conditional distribution of $A$ when the measurement has been observed up to time $t$, i.e. to have an explicit representation of $H_t := \mathbb{E}[h(A)|\mathcal{H}_t]$

for an arbitrary measurable function $h$ such that $h(A)$ is integrable. Note that by construction $H_t$ is a closed $\mathcal{H}_t$-martingale. Note also that, at least if $h(A)$ is square integrable, a conditional expectation is a best mean square approximation so that $H_t$ is the best estimate of $h(A)$ (known exactly to the cheater) for someone whose knowledge is limited to the observations. To get a hold on this conditional expectation we introduce a bounded measurable function $g(S_{t_1}, \cdots, S_{t_k})$ where $0 = t_0 < t_1 \cdots < t_k = t$ and use the general formula to get

$$\mathbb{E}\big[h(A)g(S_{t_1}, \cdots, S_{t_k})\big] = \int d\mu_0(\alpha)h(\alpha)\mathbb{E}\Big[g(B_{t_1}, \cdots, B_{t_k})e^{\alpha B_t - \alpha^2 t/2}\Big],$$

and for each $\beta \in \mathbb{R}$

$$\mathbb{E}\left[\frac{e^{\beta S_t - \beta^2 t/2}}{\int d\mu_0(\gamma)e^{\gamma S_t - \gamma^2 t/2}}g(S_{t_1}, \cdots, S_{t_k})\right] = $$
$$\int d\mu_0(\alpha)\mathbb{E}\left[\frac{e^{\beta B_t - \beta^2 t/2}}{\int d\mu_0(\gamma)e^{\gamma B_t - \gamma^2 t/2}}g(B_{t_1}, \cdots, B_{t_k})e^{\alpha B_t - \alpha^2 t/2}\right],$$

which simplifies to

$$\mathbb{E}\left[\frac{e^{\beta S_t - \beta^2 t/2}}{\int d\mu_0(\gamma)e^{\gamma S_t - \gamma^2 t/2}}g(S_{t_1}, \cdots, S_{t_k})\right] = \mathbb{E}\Big[e^{\beta B_t - \beta^2 t/2}g(B_{t_1}, \cdots, B_{t_k})\Big].$$

Integrating this identity against $\int d\mu_0(\beta)h(\beta)$ and comparing with eq.(30) we get

$$\mathbb{E}\left[\frac{\int d\mu_0(\beta)h(\beta)e^{\beta S_t - \beta^2 t/2}}{\int d\mu_0(\gamma)e^{\gamma S_t - \gamma^2 t/2}}g(B_{t_1}, \cdots, B_{t_k})\right] = \mathbb{E}\big[h(A)g(S_{t_1}, \cdots, S_{t_k})\big].$$

As $\frac{\int d\mu_0(\beta)h(\beta)e^{\beta S_t - \beta^2 t/2}}{\int d\mu_0(\beta)e^{\beta S_t - \beta^2 t/2}}$ is $\mathcal{H}_t$-measurable and $g$ is arbitrary, we have obtained our major result, an explicit representation for the closed $\mathcal{H}_t$-martingale $H_t$:

$$H_t = \mathbb{E}[h(A)|\mathcal{H}_t] = \frac{\int d\mu_0(\beta)h(\beta)e^{\beta S_t - \beta^2 t/2}}{\int d\mu_0(\beta)e^{\beta S_t - \beta^2 t/2}}.$$

This can be rephrased by saying that the measure $d\mu_0$ conditional on $\mathcal{H}_t$ is the measure (in fact a measure-valued $\mathcal{H}_t$-martingale)

$$d\mu_t(\alpha) := d\mu_0(\alpha)\frac{e^{\alpha S_t - \alpha^2 t/2}}{\int d\mu_0(\beta)e^{\beta S_t - \beta^2 t/2}}.$$

As emphasized in the main text, the same expression for $d\mu_t$ holds for the diagonal of the density matrix at time $t$ as a functional of the diagonal of the density matrix at time $0$ when the results of the measurements have been taken into account. That this must be the case is strongly supported by the discrete time counterpart recalled above.

What remains to be deciphered is how $S_t$ can be analyzed from the point of view of stochastic processes under the filtration $\mathcal{H}_t$. The general formula for the joint distribution of the random variable $A$ and the process $S_t$ leads easily to

$$\mathbb{E}[S_t - S_s|\mathcal{H}_s] = (t - s)\mathbb{E}[A|\mathcal{H}_s] \text{ for } 0 \leq s \leq t,$$

so that the process $W_t := S_t - \int_0^t \mathbb{E}(A|\mathcal{H}_s)\,ds$ is an $\mathcal{H}_t$-martingale. This can also be checked as follows. As $\mathcal{H}_t \subset \mathcal{G}_t$ and $B_t$ is a $\mathcal{G}_t$-martingale, the process

$$S_t - \mathbb{E}[A|\mathcal{H}_t]\,t = \mathbb{E}[S_t - At|\mathcal{H}_t] = \mathbb{E}[B_t|\mathcal{H}_t]$$

is an $\mathcal{H}_t$-martingale, and we are left to check that the process $\mathbb{E}[A|\mathcal{H}_t] t - \int_0^t \mathbb{E}[A|\mathcal{H}_s] ds$ is an $\mathcal{H}_t$-martingale. This is easy either by formal manipulations of conditional expectations or by integration by parts:

$$\mathbb{E}[A|\mathcal{H}_t] t - \int_0^t \mathbb{E}[A|\mathcal{H}_s] ds = \int_0^t s \, d\mathbb{E}[A|\mathcal{H}_s],$$

which is a martingale as the stochastic integral of an adapted (in fact deterministic) integrand $s$ against the martingale integrator $d\mathbb{E}[A|\mathcal{H}_s]$.

The quadratic variation of $S_t$ (which has continuous trajectories) is $dS_t^2 = dB_t^2 = dt$ and

$$\int_0^t \mathbb{E}[A|\mathcal{H}_s] ds = \int_0^t \frac{\int d\mu_0(\beta) h(\beta) e^{\beta S_s - \beta^2 s/2}}{\int d\mu_0(\beta) e^{\beta S_s - \beta^2 s/2}} \, ds$$

is a finite variation process with continuous trajectories. Hence $W_t := S_t - \int_0^t \mathbb{E}[A|\mathcal{H}_s] ds$ is an $\mathcal{H}_t$-martingale with continuous trajectories and quadratic variation $dW_t^2 = dS_t^2 = dt$. By Lévy's characterization theorem, $W_t$ is an $\mathcal{H}_t$ Brownian motion. Thus, from the point of view of the observer, the signal $S_t$ decomposes as an $\mathcal{H}_t$-semimartingale

$$S_t = W_t + \int_0^t \mathbb{E}[A|\mathcal{H}_s] ds.$$

With some tedious manipulations of the general formula allowing to go back and forth between expectations of the observation process $S_t$ and the Brownian motion $B_t$ we could do without Lévy's characterization theorem, i.e. get an explicit formula for the finite dimensional distributions of $W_t$ and recognize those of a Brownian motion. It is worth noticing that $\mathbb{E}[B_t|\mathcal{H}_t]$ which is an $\mathcal{H}_t$-martingale with continuous trajectories is not a Brownian motion.

Let us note that setting $Z(x,t) := \int d\mu_0(\beta) e^{\beta x - \beta^2 t/2}$, so that $Z_t := Z(S_t, t)$ is the normalization factor for $\mu_t$, we obtain $\mathbb{E}[A|\mathcal{H}_t] = (\partial_x \log Z)(S_t, t)$, which leads to the following form for the stochastic differential equation for $S_t$ as an $\mathcal{H}_t$-semi-martingale

$$dS_t = dW_t + (\partial_x \log Z)(S_t, t) dt = dW_t + dt \left( \int d\mu_t(\alpha) \alpha \right).$$

The first expression of the drift term is typical for a so-called $h$-transform and points to a systematic (but less direct and less elementary) derivation of the above results via Girsanov's theorem. The second expresses the instantaneous drift term as the average of the observable $A$ at time $t$ (i.e. in a state described by a density matrix whose diagonal in the pointer states basis is the measure $d\mu_t$).

One checks easily that $\mathbb{E}[h(A)|\mathcal{H}_t]$, which is automatically an $\mathcal{H}_t$-martingale, satisfies

$$d\mathbb{E}[h(A)|\mathcal{H}_t] = (\mathbb{E}[Ah(A)|\mathcal{H}_t] - \mathbb{E}[h(A)|\mathcal{H}_t] \mathbb{E}[A|\mathcal{H}_t]) \, dW_t.$$

The combinatorics ensuring the absence of $dt$ terms is embodied in the relation $\left( \partial_t + \frac{1}{2} \partial_x^2 \right) e^{\beta x - \beta^2 t/2} = 0$ valid for every $\beta$. This leads to

$$d\mathbb{E}[B_t|\mathcal{H}_t] = \left( 1 - t(\mathbb{E}[A^2|\mathcal{H}_t] - \mathbb{E}[A|\mathcal{H}_t]^2) \right) dW_t,$$

where the conditional variance on the right-hand side can be rewritten as

$$\mathbb{E}[A^2|\mathcal{H}_t] - \mathbb{E}[A|\mathcal{H}_t]^2 = \frac{\int d\mu_0(\alpha) d\mu_0(\beta) (\alpha - \beta)^2 e^{\alpha S_t - \alpha^2 t/2} e^{\beta S_t - \beta^2 t/2}}{2 \int d\mu_0(\alpha) d\mu_0(\beta) e^{\alpha S_t - \alpha^2 t/2} e^{\beta S_t - \beta^2 t/2}}$$

$$= \int d\mu_t(\alpha) d\mu_t(\beta) \frac{(\alpha - \beta)^2}{2}.$$

At large times this conditional variance vanishes and $d\mathbb{E}[B_t|\mathcal{H}_t]$ approaches the Brownian motion increment $dW_t$.

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
