# Peer review of "Monitoring continuous spectrum observables: the strong measurement limit"

_SciPost Physics, doi:SciPost Phys. 5, 037 (2018)_

## Round 2 · Referee Report · Martin Fraas (Referee 1) · 2018-8-7

Strengths

1- An important problem within the area. 2- Clear exposition. 3- Timely results that will generate further discussion of the topic.

Weaknesses

1-Discussion of relevance of the models / the scaling limits considered is missing.

Report

The paper develops further the perturbation theory for non-demolition measurements in quantum mechanics. The case of discrete observables is now reasonably well understood. In this article the authors do a very natural (and also important) step to study the theory for observables with continuous spectrum. This step is not only technically more difficult but also presents a conceptual challenge due to the non-existence of long time limit of posterior states for the unperturbed dynamics.

\\

The authors study two situations: (a) Dissipative perturbation, (b) Hamiltonian perturbation. In both cases, the authors provide a convincing analysis leading to the leading order effective dynamics (in appropriate rescaling). In the case (a) they prove a theorem covering large class of systems. In the case (b) they analyze a particular example in details and give good arguments for a general conjecture. Both results are of high quality and important for the current developments in the area.

\\

The paper is well written. For example, I really like the metaphor of the cheater that is nicely connected to different filterings and conditional expectation values.

\\

I was wondering about the following two questions while reading the article. The first one is how much of the results can be inferred from the perturbation theory of Lindbladians (e.g. Appendix of https://arxiv.org/pdf/1512.05801.pdf) without any discussion of the unraveled equations. (Or maybe more precisely how much this can be decoupled). For example in the case (a) in the Heisenberg picture the Lindbladian associated to the average evolution in Eq.(9) has for $D=0$ a kernel consisting of any functions of the position operator $f(X)$. The perturbation theory for Lindbaldians then gives that for large $\gamma$ (in the leading order) the evolution inside the kernel is generated by the projected perturbation, which is exactly the generator found above Eq.(14). Similarly (I believe) a semi classically analysis of the Lindbaldian in (b) would lead to the generator (and manifold) of the Markov process in (25), (26).

The second question is how relevant are the models and the scaling limits? For example is the model in 4.1. a good description of some particle detector? If yes, is the classical limit, the relevant physics limit for the detector?

Requested changes

1-Should there be hat on rho in Eq. (10)? 2-What is the meaning of bar on Y in Eq.(12)? 3-Definition of $\mathbb{E}_{\mu_0}$ is missing. 4-It took me some time to decipher the meaning of $\mathcal{D} \cdot$ maybe some more explanation of which operators acts on what objects might help. 5- $\Pi_t^{\alpha}$ below Eq.(14) should be probably $\Pi_t^{\gamma}$. 6-The discussion of reverse martingales is intriguing but cryptic. For example can you provide a reference for the claim that "reverse martingales are the right generalization for means of partial sums ...". 7-What is the hat on psi below equation (20) 8-On page 19 above third displayed Eq. from the bottom, what does "comparing" refer to? Also in that equation should B be S?

---

## Round 2 · Referee Report · Anonymous (Referee 2) · 2018-8-14

Strengths

  1. The paper is mostly extremely clear and pedagogical.
  2. The new results are clearly highlighted and exemplified.
  3. A conjecture is made for the limit of strong monitoring for a particle in a general potential, which should motivate further mathematical work.

Weaknesses

  1. Because the paper represents a synthesis of ideas from measurement theory and stochastic processes, some technical ideas are not described in sufficient detail for a reader with background in only one of these fields.

Report

This paper provides a rigorous formulation of the challenging problem of continuous monitoring of a quantum observable taking continuous values (e.g. particle position) in the strong measurement limit.

In the case of discrete observables this limit leads to a classical Markov process on the possible measurement outcomes. In the continuous case one naturally expects a diffusion-type process. This paper shows how to derive this process in a controlled way for two important cases -- where the strong monitoring competes with (1) dissipative and (2) Hamiltonian evolution. The general case of dissipative evolution is resolved, but the authors treat only the case of harmonic oscillator evolution in the Hamiltonian case, making a conjecture about strong monitoring in the presence of evolution in a smooth potential.

Requested changes

  1. The quantity $\gamma$ is not defined in Eq. (1)!

  2. A sketch of the derivation of Eq. (1) and (2) is given in Section 2. It would be a shame not to flesh this out (it would also show where $\gamma$ comes from).

  3. I'm not sure that introducing the probability space $(\Omega, \mathcal{G}, p)$ helps the clarity of the theorem stated in Section 2.1, as none of these quantities appear in the subsequent statement of the theorem (I know $\mathcal{G}$ appear in the proof: maybe it could be defined there?).

  4. In the statement of that theorem: would it correct to say that $W_t$ is a Brownian motion (dropping the $\mathcal{H}_t$-adapted?

  5. End of Section 2.2: "for any bounded function.". Perhaps should add "of compact support"?

---

## Round 3 · Author Response

We thank both referees for their positive comments on our manuscript.

— Let us first answer the first question by referee 1:

The first question asks "how much of the results can be inferred from the
perturbation theory of Lindbladians (…) without any discussion of the unraveled
equations. (…)?". The Lindblad operator codes linearly for the mean evolution of the
density matrix. This perturbative approach can of course be done but it will only
give information on the behaviour of functions linear in the density matrix. The
statement we have proved concerns the behavior of all polynomials (or convergent
series) in the density matrix, which cannot be deduced from the perturbative theory
of the Lindladian. Nevertheless, the proof we have given is close to this suggested
perturbative approach but it applies to operators acting on any polynomials in the
density matrix.

The second question asks "how relevant are the models and the scaling limits?". For
case (a), the limit is simply the limit of large information extraction rate without
any scaling of the other coupling constants. The case (b) is more delicate because
the Zeno effect takes place at large extraction rate and freezes the dynamics unless
appropriate scalings of the couplings constants is chosen, as is well known. As
recalled in the introduction, the case (b) possesses three different regimes
depending on the time scale: (a) a collapse regime, (b) a classical regime in which
the localised wave function moves in space according to classical dynamics, and (c)
a diffusive regime in which the wave function diffuses randomly. As explained in the
text (in the introduction and at the beginning of section 4), the scalings we have
chosen iare adapted to describe the cross-over from regime (b) to (c), which is
therefore within the semi-classical approximation.

Whether the model in 4.1. is a good description of some particle detector is a
question worth asking. It is well documented that the model of section 4, which is
at the basis of the theory of quantum trajectories, is a good description of the
quantum back-action induced by the monitoring of an observable with continuous
spectrum. Nevertheless, as exemplified by the famous (but yet unsolved) Mott track
problem, it remains an open question whether such quantum trajectory equations are
adapted to describe the detection of particle trajectories in say bubble chambers
(or in any other similar particle detection devices).

— Let now answer the referee 2 comments:

As pointed by the referee our paper "represents a synthesis of ideas from
measurement theory and stochastic processes". Although we have tried our best to
introduce and explain the objects and the techniques we used, we necessary had to
assume that the reader has some (basic) knowledge on both quantum measurement and
probability theory. Making this assumption is unavoidable and it is part of the
difficulties (but also part of beauty) of this scientific topics. But, besides
references [1,2] to books already included, we have added a reference to our lecture
notes on this topic.

---

## Round 3 · List of Changes

Referee 1 We have implemented all the requested changes: - points 1, 2, 3, 5 and 7 concern misprint that we corrected. - we didn’t implement point 4 because, even if there are more intrinsic formulations of the action of the operator D), there are not as explicit as the straightforward one given in the text and less useful. - point 6: we have given a more explicit formulation of the statement "reverse martingales are the right generalization for means of partial sums" and added a reference. - point 8: we have labeled the appropriate equation to which the comparison refers.

Referee 2 We have implemented all the requested changes: - point 1 concerns a misprint that we corrected. - point 2: we think giving the derivation will render the text too heavy (because it is bit long) but we have added a reference to our lecture notes on this topic. - point 3: as suggested by the referee, we change the formulation, to make simpler, and we remove the reference to an explicit probability space. - point 4: The referee is right that the theorem remains valid if « H_t adapted » is dropped. The theorem simply becomes less precise. However keeping the full statement (including H_t adapted) is crucial because H_t is the information collected by observing only the outcomes of the indirect measurements : W_t is accessible to the observer whereas B_t is'nt. We have introduced different filtrations and the main point of this theorem is to explain the interplay between these distinct filtrations. - point 5: the referee points correctly that we have to better characterise the function for which the statement is correct. We did it in the text.

---

## Editorial Decision

published